# Diagnosis of invasive pulmonary aspergillosis using metagenomic next-generation sequencing and conventional microbial tests post-COVID-19 pandemic

Ping Li,[1,2] Yuxia Chen,[3] Xuelei Cao,[1,2] Zheying Wang,[1,2] Lintao Sai,[4] Lili Wang[1,2]

**ABSTRACT** Early recognition and timely diagnosis are crucial for improving the clinical outcome of invasive pulmonary aspergillosis (IPA) patients. Metagenomic next-generation sequencing (mNGS) shows immense advantages in identifying responsible complex pathogens, especially with the gradual ease of COVID-19 control policies in China since 2022. A total of 327 patients with suspected *Aspergillus* infection in non-neutropenic populations were enrolled in the current study. The diagnostic efficacy with mNGS and conventional microbial tests (CMTs) in suspected IPA patients was assessed, and the incidence and risk factors for *Aspergillus* infection were also investigated. mNGS exhibited excellent performance in detecting *Aspergillus*. The sensitivity of mNGS (80.58%) was superior to that of CMTs, as demonstrated by comparisons with smears (22.30%, $P < 0.001$), culture (30.94%, $P < 0.001$), serum GM (22.62%, $P < 0.001$), BALF GM (55.40%, $P < 0.001$), and combined CMTs (61.87%, $P < 0.001$). The results of mNGS caused a direct shift in the management of 212 (64.8%) positive effect patients, making a clear diagnosis and instructing antifungal therapy. Notably, in addition to the common risk factors, the patients with a history of COVID-19 infection were more prone to IPA. The occurrence of IPA increased significantly with the gradual ease of COVID-19 control policies (47.62% vs 30.21%, $P = 0.004$). Meanwhile, mixed infections were commonly observed in IPA patients, with *Human gammaherpesvirus* and *Acinetobacter baumannii* being the most common co-pathogens. Our study demonstrated that mNGS might present a feasible and remarkably sensitive approach for detecting *Aspergillus*, thereby serving as a valuable auxiliary tool for CMTs.

**IMPORTANCE** Our study is the first to focus on *Aspergillus* infection after the COVID-19 pandemic and find that (i) mNGS is a feasible and highly sensitive method for detecting *Aspergillus* post-COVID-19 pandemic, thereby serving as a valuable auxiliary tool for CMTs. (ii) mNGS has the potential to revolutionize the management of fungal infections. (iii) The history of COVID-19 infection is an independent risk factor for IPA. Identification of this risk factor for IPA may raise clinical attention and require careful follow-up of high-risk individuals post-COVID-19 infection. (iv) Mixed infections were commonly observed in IPA patients, with *Human gammaherpesvirus* and *Acinetobacter baumannii* being the most common co-pathogens.

**KEYWORDS** invasive pulmonary aspergillosis (IPA), metagenomic next-generation sequencing (mNGS), conventional microbial tests (CMTs), diagnosis, post-COVID-19 pandemic

Invasive pulmonary aspergillosis (IPA) is an opportunistic infection caused by the ubiquitous fungus *Aspergillus*, frequently occurs in immunocompromised individuals, especially those with prolonged neutropenia, long-term use of corticosteroids, recipients

Address correspondence to Lili Wang, liliwang@sdu.edu.cn.

The authors declare no conflict of interest.

of hematopoietic stem cell or solid organ transplants, and may cause higher life-threatening infection (1, 2). In fact, due to the extensive use of antibiotics and corticosteroids, the incidence of IPA has increased greatly in recent years, especially with the gradual easing of COVID-19 control policies in China since 2022 (3, 4).

Definite diagnosis of IPA primarily relies on pathologic evidence combined with risk factors, clinical features, and microbiological evidence (5). Pulmonary histopathological examination remains the gold standard for IPA diagnosis, but the fact that many patients do not tolerate tissue biopsy has prevented wide clinical usage (6). *Aspergillus* culture is currently the most used diagnostic tool, but it is time-consuming and has low sensitivity (7). Although smear by microscopy can identify *Aspergillus* hyphae more quickly than culture, it requires a heavy pathogen burden and experienced microbiologists to ensure sensitive detection. Poor applicability of traditional host factors, intolerance to biopsy, low sensitivity of culture, and absence of typical clinical and radiological findings are limiting factors for early pulmonary aspergillosis diagnosis (8). Serum and bronchoalveolar lavage fluid (BALF) galactomannan antigen (GM) tests are recognized as an early biomarker of *Aspergillus*, widely applied in aspergillosis diagnosis (9, 10). However, the sensitivity of GM in serum samples from non-neutropenic hosts is limited due to low *Aspergillus* loads in the lesions and less chance of antigen appearing in the bloodstream (11). BALF GM test is superior to serum GM in the diagnosis of IPA (12), but the results may be influenced by standardization of the sampling, antimicrobial treatment, and various other factors.

Metagenomics next-generation sequencing (mNGS) is an unbiased pathogen detection and molecular technology of nucleic acid sequencing, which has been considered as a promising microbial identification technology in infectious diseases (13). The application of mNGS shows immense advantages over traditional detection in identifying responsible pathogens (14). Due to the difficulty of DNA extraction from the thick polysaccharide cell wall and the relatively low fungal load in BALF, identifying filamentous molds, such as *Aspergillus spp.*, through mNGS remains challenging (15, 16). To date, the diagnostic performance of mNGS for the detection of lung *Aspergillus* infection is still lacking strong evidence.

Recent studies have indicated that the incidence of *Aspergillus* infection increased after COVID-19 infection (17, 18). COVID-19-associated pulmonary aspergillosis must be considered a serious and potentially life-threatening complication in patients with severe COVID-19 receiving immunosuppressive treatment. From a pathophysiological point of view, viral infections favor the development of fungal infections. A damaged respiratory epithelium, dysfunctional mucociliary clearance, the suppression of cellular immunity, and the alteration of phagocytic activity were demonstrated to be potential explanations for the coexistence of viral and fungal infections (19). After the COVID-19 pandemic, the high incidence of *Aspergillus* infection was not limited to people with severe COVID-19 illness. Aberrant pulmonary architecture and functioning have been described in many recovered COVID-19 patients (20). It was still necessary to study the overall incidence and risk factors for *Aspergillus* infection, especially occurring post-COVID-19.

In this retrospective study, we strive to compare the diagnostic efficacy with mNGS and conventional microbial tests (CMTs) in pneumonia patients with suspected *Aspergillus* infection, assess the clinical effects of mNGS results, and investigate the incidence and risk factors for *Aspergillus* infection occurring post-COVID-19, which might be important to help physicians with rational therapy and possible prevention strategies.

## MATERIALS AND METHODS

### Study design and subjects

A total of 347 patients with suspected *Aspergillus* infection in non-neutropenic populations were enrolled at Qilu Hospital of Shandong University from October 2021 to June 2023. In this study, the timeframe from 1 October 2021 to 7 December 2022 was defined as the pre-COVID-19 pandemic era, while 8 December 2022 to 30 June

2023 was categorized as the post-COVID-19 pandemic era. This division aligns with the gradual easing of COVID-19 control policies in China since 7 December 2022. The inclusion criteria of "non-neutropenic suspected IPA" was identified as (i) patients had risk factors for IPA, including but not limited to recipients of hematopoietic stem cell or solid organ transplants, prolonged use of corticosteroids, structurally destructive lung diseases (e.g. chronic obstructive pulmonary disease [COPD], bronchiectasis), severe influenza, diabetes, and malnutrition (21). (ii) patients failed to receive empirical antimicrobial therapy; (iii) abnormal chest radiographic images suggestive of pulmonary aspergillosis; (iv) bronchoscopy revealed tracheobronchial pseudomembranous, ulcers, nodules, plaques, or eschars; (v) peripheral blood neutrophil count was $>0.5 \times 10^9$/L. The following exclusion criteria were as follows: (i) age <18 years old; (ii) medical records were incomplete.

Clinical information was extracted from the electronic medical records using a standardized data collection form, which included demographic information, clinical data, laboratory findings, CT of the chest, antifungal treatments, and outcomes. All the data were reviewed by two investigators independently to verify data accuracy. All the enrolled patients were followed up by telephone to inquire about survival or date of death.

## Diagnostic criteria for IPA

The histopathological evidence of *Aspergillus* hyphae in lung biopsy specimens was considered proven IPA. The probable IPA needs evidence of at least one host factor, radiologic features, and mycologic evidence according to the guidelines of the 2020 European Organization for Research and Treatment of Cancer and the Mycoses Study Group Education and Research Consortium (EORTC/MSGERC) (5). The details were as follows: (i) immunocompromised patients (including hematologic malignancy, hematopoietic stem cell or solid organ transplants, prolonged use of corticosteroids, taken antirheumatic drugs, biological immunomodulators or immunosuppressants); (ii) radiologic features were dense, well-circumscribed lesions with or without a halo sign, an air crescent sign, cavity or wedge-shaped, and segmental or lobar consolidation; (iii) when any of the following conditions was met, it would be used as mycological evidence: (i) GM test ≥0.5 in serum or ≥1.0 in BALF; (ii) positive smear or culture results (qualified specimen from sputum, BALF, or bronchial brush). Possible IPA met at least one host factor and clinical feature. All these enrolled probable cases were further evaluated by the response to antifungal treatment.

The final determination of causative pathogens, colonization, or contamination was based on clinical comprehensive diagnostic criteria, which was established by two senior specialists after independently reviewing the electronic medical records of each patient, based on clinical symptoms, radiologic features, microbiologic tests, and treatment responses. Any differences were resolved through in-depth discussion and consensus.

## Microbiological testing and mNGS

BALF specimens were collected by experienced physicians through bronchoscopy in accordance with standard procedures (22). A portion of extracted BALF was sent for culture, smear microscopy, and GM tests within 2 h, while the remaining specimens were sent for mNGS analysis. The *Aspergillus* GM antigen assay in BALF and serum was quantified using Platelia *Aspergillus* Ag (Bio-Rad, USA), following the manufacturer's instructions. The process of mNGS analysis included deoxyribonucleic acid (DNA) extraction, library construction, high-throughput sequencing, and bioinformatics analysis in the BGISEQ-200 platform. Detailed procedures are given in the Supplement.

## Statistical analyses

Continuous variables were expressed as median (IQR) and compared with the Mann-Whitney U test. Categorical variables, shown as frequencies and percentages [n (%)],

were compared by χ2 test or Fisher's exact test. The 2 × 2 contingency tables were established to determine sensitivity, specificity, positive predictive value (PPV), and negative predictive value (NPV). Wilson's method was used to calculate 95% confidence intervals (CIs) for these proportions. The McNemar test was used for comparisons of the diagnostic performance of CMTs and mNGS. Univariate and multivariate logistic regression models were used to explore the risk factors associated with IPA diagnoses. The Kaplan-Meier method was used to compare the 28-days overall survival. The SPSS 25.0 software was used for data analysis, and $P < 0.05$ was considered statistically significant.

## RESULTS

### Patients' recruitment and clinical characteristics

A total of 347 patients with suspected IPA were enrolled in our study. After excluding three patients with unmatched pairs of mNGS and CMTs, 12 patients who refused to undergo bronchoscopy to obtain BALF, and five with incomplete medical records, 327 patients were finally eligible for final analysis in this study. Of these patients, 139 patients were diagnosed with IPA, consisting of 11 proven patients, 74 probable patients, and 54 possible patients. A total of 188 patients were finally diagnosed with non-IPA based on the clinical composite diagnosis. (Fig. 1)

The primary characteristics of all the eligible patients are presented in Table 1. The median ages of these two groups were similar. There were no significant differences in most symptoms (including cough, expectoration, fever, dyspnea, and hemoptysis) between the IPA and non-IPA groups. However, IPA patients had a higher incidence of coma or drug sedation (28.06% vs 18.62%, $P = 0.044$) compared with the non-IPA group. Meanwhile, we observed that patients with a history of COVID-19 infection, interstitial lung disease, diabetes mellitus, organ transplantation, and systemic corticosteroids were preferentially suffering from IPA ($P < 0.01$). The typical radiological features of IPA on chest CT were infrequent. In comparison with the non-IPA group, halo sign (7.91% vs 1.06%, $P = 0.002$) and cavities (21.58% vs 13.30%, $P = 0.048$) were more prevalent in the

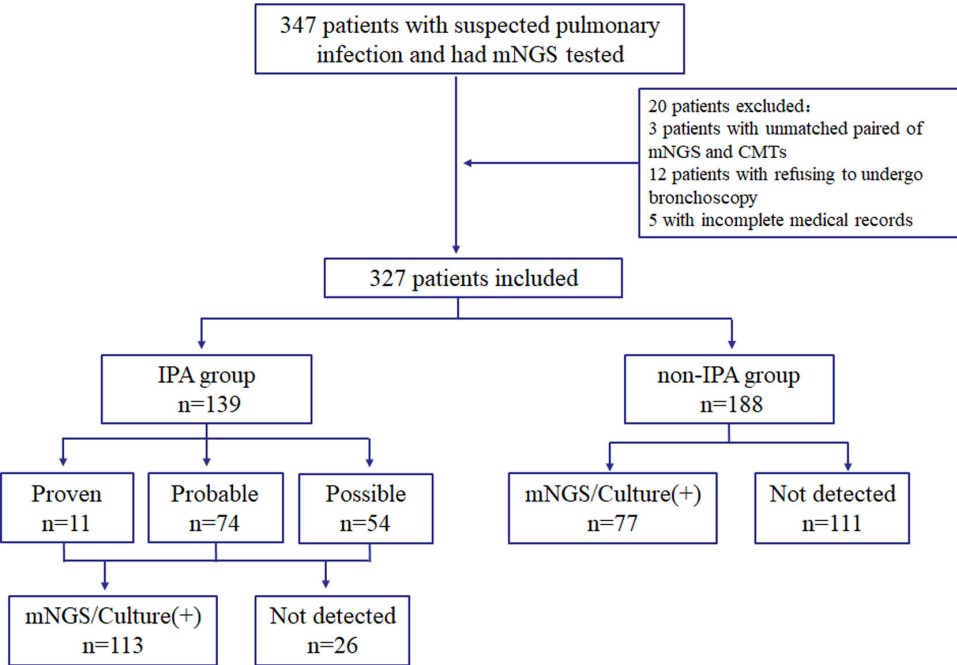

**FIG 1** Study workflow. Schedule of all enrolled patients. 347 suspected patients were initially enrolled in this retrospective study, and 327 patients were finally included in the analysis. There were 139 patients diagnosed as invasive pulmonary aspergillosis (IPA) and 188 patients diagnosed as non-IPA.

**TABLE 1** Baseline data of patients in IPA and non–IPA groups[a,b]

| Characteristics | IPA (n = 139) | non–IPA (n = 188) | P–value |
|---|---|---|---|
| Male, n (%) | 91 (65.47%) | 135 (71.81%) | 0.220 |
| Age (year), median (IQR) | 62 (53–70) | 67 (57–73) | 0.008[**] |
| Symptoms, n (%) | | | |
| Cough | 62 (44.60%) | 78 (41.49%) | 0.574 |
| Expectoration | 48 (34.53%) | 68 (36.17%) | 0.760 |
| Fever (body temperature >37.1°C) | 61 (43.88%) | 93 (49.47%) | 0.317 |
| Dyspnea | 47 (33.81%) | 54 (28.72%) | 0.325 |
| Hemoptysis | 5 (3.60%) | 7 (3.72%) | 0.952 |
| Coma or drug sedation | 39 (28.06%) | 35 (18.62%) | 0.044[*] |
| Underlying pulmonary diseases, n (%) | | | |
| COPD | 10 (7.19%) | 16 (8.51%) | 0.664 |
| Bronchiectasis | 4 (2.88%) | 10 (5.32%) | 0.267 |
| Interstitial lung disease | 20 (14.39%) | 9 (4.79%) | 0.003[**] |
| Obsolete pulmonary tuberculosis | 3 (2.16%) | 5 (2.66%) | 0.772 |
| Lung cancer | 5 (3.60%) | 3 (1.60%) | 0.247 |
| Bronchial asthma | 4 (2.88%) | 4 (2.13%) | 0.664 |
| Extrapulmonary diseases, n (%) | | | |
| Diabetes mellitus | 44 (31.65%) | 33 (17.55%) | 0.003[**] |
| Hypertension | 56 (40.29%) | 71 (37.77%) | 0.644 |
| Cardio–cerebrovascular diseases | 42 (30.22%) | 53 (28.19%) | 0.690 |
| Autoimmune diseases | 11 (7.91%) | 10 (5.32%) | 0.344 |
| Hematologic malignancy | 19 (13.67%) | 15 (7.98%) | 0.096 |
| Active solid tumor | 13 (9.35%) | 11 (5.85%) | 0.230 |
| Organ transplantation | 11 (7.91%) | 3 (1.60%) | 0.005[**] |
| Systemic corticosteroids | 17 (12.23%) | 9 (4.79%) | 0.014[*] |
| History of COVID–19 infection | 110 (79.14%) | 121 (64.36%) | 0.004[**] |
| Radiological findings, n (%) | | | |
| Ground glass opacity | 17 (12.23%) | 20 (10.64%) | 0.653 |
| High–density film | 124 (89.21%) | 170 (90.43%) | 0.718 |
| Diffuse Interstitial infiltration | 19 (13.67%) | 20 (10.64%) | 0.403 |
| Nodules | 39 (28.06%) | 66 (35.11%) | 0.177 |
| Halo sign | 11 (7.91%) | 2 (1.06%) | 0.002[**] |
| Cavitation sign | 30 (21.58%) | 25 (13.30%) | 0.048[*] |
| Consolidation | 19 (13.67%) | 27 (14.36%) | 0.859 |
| Bronchoscopy findings, n (%) | | | |
| Tracheobronchial ulcers, nodules, pseudomembranes, plaques, or eschars | 31 (22.30%) | 13 (6.91%) | 0.000[***] |
| Length of stay in hospital (days), median (IQR) | 17 (13–25) | 18 (12–30) | 0.597 |
| 28 days mortality, n (%) | 55 (39.57%) | 28 (14.89%) | 0.000[***] |

[a]IPA: invasive pulmonary aspergillosis; COPD: chronic obstructive pulmonary disease; IQR: interquartile ranges.
[b]*, $P<0.05$; **, $P<0.01$; ***, $P<0.001$.

IPA group based on a retrospective review of radiological images. Compared with the non-IPA group, tracheobronchial ulcers, nodules, pseudomembranes, plaques, or eschars were more common in the bronchoscopy findings of IPA patients (22.30% vs 6.91%, $P < 0.001$). In these patients, no significant difference in the number of days in hospital was observed between the IPA and non-IPA groups (17 days vs 18 days; $P = 0.597$). The 28-day mortality of IPA patients was higher than that of the non-IPA group (39.57% vs 14.89%, $P < 0.001$).

## Comparison of diagnostic performance between mNGS and CMTs

Among the 139 IPA patients of our study, approximately 35.25% (49/139) were positive both in mNGS and traditional tests, 18.71% (26/139) were negative both in the mNGS and traditional tests, mNGS was only positive in 45.32% (63/139) of the patients, and one patient was only positive with the traditional test (Fig. 2A). Meanwhile, a total of 157 strains of *Aspergillus sp*. were detected in IPA patients using mNGS, including 93 *Aspergillus fumigatus*, 43 *Aspergillus flavus*, 5 *Aspergillus niger*, 12 *Aspergillus terreus*, 2 *Aspergillus tubingensis*, and 2 *Aspergillus nidulans*. However, only 35 *Aspergillus fumigatus*, 3 *Aspergillus flavus*, 2 *Aspergillus niger,* and 2 *Aspergillus terreus* were identified with traditional culture (Fig. 2B). It was worth noting that in one patient, the *Aspergillus* culture result was *Aspergillus niger*, while no *Aspergillus* was detected by mNGS.

As provided in Table 2, mNGS identified *Aspergillus* specific sequences in 112 of 139 patients with IPA and 77 of 188 patients without IPA, resulting in a sensitivity of 80.58% (95% CI, 72.82%–86.60%), specificity of 59.04% (95% CI, 51.63%–66.07%), PPV of 59.26% (95% CI, 51.87%–66.27%), NPV of 80.43% (95% CI, 72.63%–86.49%), and accuracy of 68.20% (95% CI, 62.81%–73.16%). Then, we evaluated the diagnostic efficacy of a single CMT. In the IPA group, smear, culture, serum GM, and BALF GM were positive in 31, 43, 37, and 77 cases, respectively, while in the non-IPA group, there were 4, 4, 17, and 14 cases, respectively. The sensitivity, specificity, PPV, and NPV of culture were 30.94% (95% CI, 23.53%–39.42%), 97.87% (95% CI, 94.29%–99.31%), 91.49% (95% CI, 78.73%–97.24%), 65.71% (95% CI, 59.79%–71.19%), 69.42% (95% CI, 64.07%–74.31%), respectively, better than smear and serum GM. Meanwhile, the sensitivity of mNGS was superior to that of CMTs, as demonstrated by comparisons with smears (22.30%, $P <$ 0.001), culture (30.94%, $P <$ 0.001), serum GM (22.62%, $P <$ 0.001), BALF GM (55.40%, $P <$ 0.001), and combined CMTs (61.87%, $P <$ 0.001). The sensitivity of BALF GM was suboptimal, better than that of serum GM. There was a statistically significant difference in the level of serum/BALF GM between the IPA and non-IPA groups (Fig. 3), which indicated that the serum/BALF GM assay might be an optional diagnostic assay for IPA. Furthermore, we evaluated the diagnostic efficacy of a variety of routine microbial tests. The combination of multiple CMTs determined positive results in 86 of 139 IPA patients and 32 of 188 non-IPA patients, and the sensitivity, specificity, PPV, NPV, and accuracy were 61.87% (95% CI: 53.22%–69.86%), 82.98% (95% CI: 76.67%–87.91%), 72.88% (95% CI: 63.78%–80.46%), 74.64% (95% CI: 68.08%–80.27%), 74.00% (95% CI: 68.84%–78.61%), respectively. It showed higher sensitivity and specificity when compared with those of BALF GM ($P <$ 0.01).

## Clinical effects of mNGS results on IPA diagnosis and management

When evaluating the clinical effects of mNGS results on IPA diagnosis and management, we found that they had a positive or no effect in 212 (64.8%) and 104 (31.8%) patients

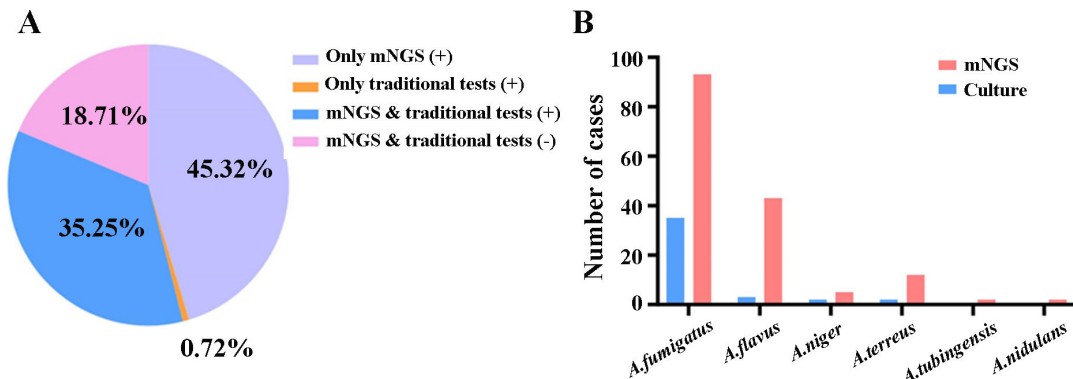

**FIG 2** Comparison of diagnostic performance between mNGS and traditional tests (including culture and smear). (A) The concordance in the positive between mNGS and traditional tests in the diagnosis of IPA. (B) The comparison of 157 *Aspergillus* strains detected by mNGS and culture in 139 IPA patients.

**TABLE 2** Diagnostic performance of mNGS and CMTs for invasive pulmonary aspergillosis[a]

| Methods | IPA $n = 139$ | Non–IPA $n = 188$ | Sensitivity% (95% CI) | Specificity% (95% CI) | PPV% (95% CI) | NPV% (95% CI) | Accuracy% (95% CI) |
|---|---|---|---|---|---|---|---|
| mNGS | | | | | | | |
| Pos | 112 | 77 | 80.58 | 59.04 | 59.26 | 80.43 | 68.20 |
| Neg | 27 | 111 | (72.82–86.60) | (51.63–66.07) | (51.87–66.27) | (72.63–86.49) | (62.81–73.16) |
| Smear | | | | | | | |
| Pos | 31 | 4 | 22.30[b] | 97.87 | 88.57 | 63.01 | 65.75 |
| Neg | 108 | 184 | (15.87–30.30) | (94.29–99.31) | (72.32–96.27) | (57.16–68.51) | (60.29–70.83) |
| Culture | | | | | | | |
| Pos | 43 | 4 | 30.94[b] | 97.87 | 91.49 | 65.71 | 69.42 |
| Neg | 96 | 184 | (23.53–39.42) | (94.29–99.31) | (78.73–97.24) | (59.79–71.19) | (64.07–74.31) |
| Serum GM ≥ 0.5 | | | | | | | |
| Pos | 37 | 17 | 26.62[b] | 90.96 | 68.52 | 62.64 | 63.61 |
| Neg | 102 | 171 | (19.65–34.90) | (85.69–94.49) | (54.31–80.10) | (56.58–68.34) | (58.11–68.79) |
| BALF GM ≥ 1.0 | | | | | | | |
| Pos | 77 | 14 | 55.40[b] | 92.55 | 84.62 | 73.73 | 76.76 |
| Neg | 62 | 174 | (46.75–63.75) | (87.57–95.71) | (75.20–91.04) | (67.54–79.13) | (71.73–81.15) |
| Combined CMTs | | | | | | | |
| Pos | 86 | 32 | 61.87[b,c] | 82.98 | 72.88 | 74.64 | 74.00 |
| Neg | 53 | 156 | (53.22–69.86) | (76.67–87.91) | (63.78–80.46) | (68.08–80.27) | (68.84–78.61) |

[a]mNGS: metagenomic next–generation sequencing; GM: galactomannan; BALF: bronchoalveolar lavage fluid; CMTs: conventional microbiological tests; Pos: positive; Neg: negative; PPV: positive predictive value; NPV: negative predictive value; CI: confidence interval.
[b]The difference was significant between CMTs and mNGS based on the McNemar test (P<0.001).
[c]The difference was significant between combined CMTs and BALF GM based on the McNemar test (P<0.01).

separately, while a negative effect was reported in 11 patients (3.4%). The results of mNGS played a definite role in diagnosis and guidance of medication in 212 patients. Of these, 113 patients were in the IPA group, and 99 patients were in the non-IPA group. In the light of treatment, mNGS results caused a direct shift in the management of 212 positive effect patients. Among the 113 IPA patients, 73 patients were given voriconazole immediately based on the mNGS results, and 40 patients received other antifungal treatment in addition to voriconazole. Among the 99 non-IPA patients, 54 patients were given voriconazole based on the mNGS results, 13 patients were given other antifungal treatments in addition to voriconazole, 17 patients were treated with amphotericin B, sulfamethoxazole, and echinocandins due to the detection of *Rhizomucor*, *Rhizopus*, *Pneumocystis jirovecii,* and *Candida*, and 15 patients stopped voriconazole due to mNGS results. However, mNGS failed to identify any *Aspergillus* in 104 patients

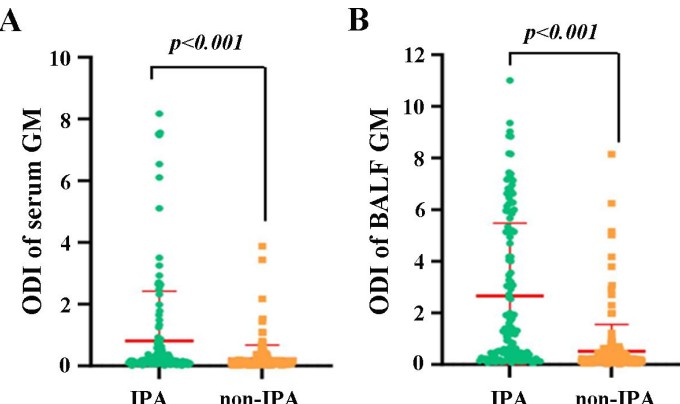

**FIG 3** The comparison of the galactomannan antigen (GM) levels between the IPA group and the non-IPA group. (A) The ODI of serum GM in the IPA and non-IPA group. (B) The ODI of BALF in the IPA and non-IPA group. ODI, optical density index; BALF, bronchoalveolar lavage fluid.

with suspected *Aspergillus* infection, but empirical voriconazole therapy continued to be administered clinically. Notably, there were 11 cases of mNGS results with negative impact; its consequences in these patients were considered contaminated or insubstantial (Table 3).

To further analyze whether mNGS diagnosis has a positive impact on patient prognosis, all patients were divided into two groups based on the detection of *Aspergillus* by mNGS: mNGS (+) group and mNGS (-) group, containing 189 and 138 patients, respectively. The 28-days overall survival (OS) in the mNGS (+) group was significantly lower than that in the mNGS (-) group ($P = 0.018$) (Fig. S1), indicating that mNGS diagnosis plays a good role in suggesting a poor prognosis for patients and helping physicians to formulate possible preventive strategies in time.

## Mixed infections and co-pathogens detected by mNGS in IPA patients

Mixed infections were commonly observed in IPA patients. The numbers of cases with mixed infections detected by mNGS were summarized in Fig. 4. Viruses and bacteria were the most common co-pathogens observed in IPA patients. The most common combination modes were *Aspergillus-bacteria-virus* coinfection ($n = 34$) and *Aspergillus-virus* coinfection ($n = 23$) (Fig. 4A). The top five co-pathogens detected by mNGS included *Human gammaherpesvirus*, *Acinetobacter baumannii*, *Klebsiella*, *Rhizomucor/Rhizopus,* and *Stenotrophomonas maltophilia* (Fig. 4B) in the current study. Of note, *Rhizomucor/Rhizopus* was identified in 17 (12.2%) of 139 patients, indicating that more attention should be paid to mixed infections of *Aspergillus* and *Rhizomucor/Rhizopus* in clinic.

## Risk factors for IPA diagnosis

We assessed the baseline data of IPA and non-IPA patients using univariate logistic regression. Age, coma/drug sedation, interstitial lung disease, diabetes mellitus, organ transplantation, systemic corticosteroids, history of COVID-19 infection, halo sign, cavitation sign, and bronchoscopy findings were found to be statistically significant between the two groups, suggesting that these factors were associated with IPA diagnosis (Table 4). Covariates with a $P < 0.05$ in the univariate analysis were included in the final model of multivariable logistic regression. The results showed that interstitial

**TABLE 3** Clinical effects of mNGS results on IPA diagnosis and management of 327 patients[a]

| Clinical effect | Patients N (%) | Role of mNGS results | Patients N (%) | | Treatment changes | Patients N (%) | |
|---|---|---|---|---|---|---|---|
| | | | IPA | Non-IPA | | IPA | Non-IPA |
| Positive effect | 212 (64.8%) | Make a definite diagnosis and instruct antifungal therapy | 113 (34.6%) | 99 (30.3%) | Voriconazole was given | 73 (22.3%) | 54 (16.5%) |
| | | | | | Add echinocandins Or amphotericin B Or isavuconazole Or SMZ | 40 (12.2%) | 13 (4.0%) |
| | | | | | Chang voriconazole to SMZ Or amphotericin B Or echinocandins | 0 (0%) | 17 (5.2%) |
| | | | | | Stop voriconazole | 0 (0%) | 15 (4.6%) |
| No effect | 104 (31.8%) | No *Fungi* detected | 26 (8.0%) | 78 (23.9%) | Empirical voriconazole therapy continued | 26 (8.0%) | 75 (22.9%) |
| | | | | | No voriconazole was given | 0 (0%) | 3 (0.9%) |
| Negative effect | 11 (3.4%) | False-positive results | 0 (0%) | 11 (3.4%) | No voriconazole was given | 0 (0%) | 11 (3.4%) |

[a]SMZ: Sulfamethoxazole.

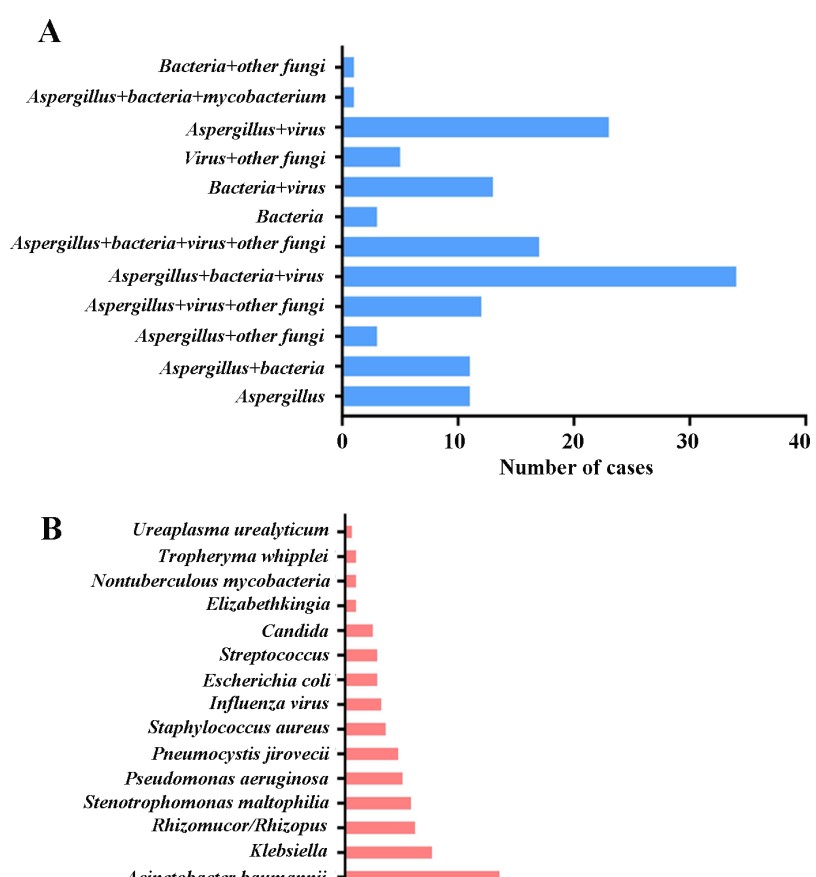

**FIG 4** Mixed infections and co-pathogens identified by mNGS in 139 IPA patients. (A) The number of IPA patients with mixed infections. (B) The number of co-pathogen infections in IPA.

lung disease (OR = 3.439, 95% CI: 1.404–8.427, $P$ = 0.007), diabetes mellitus (OR = 1.934, 95% CI: 1.086–3.443, $P$ = 0.025), organ transplantation (OR = 4.817, 95% CI: 1.191–19.481, $P$ = 0.027), systemic corticosteroids (OR = 3.135, 95% CI: 1.251–7.852, $P$ = 0.015), history of COVID-19 infection (OR = 2.120, 95% CI: 1.204–3.732, $P$ = 0.009), halo sign (OR = 5.460, 95% CI: 1.039–28.699, $P$ = 0.045), and bronchoscopy findings (OR = 3.012, 95% CI: 1.430–6.343, $P$ = 0.004) were independent risk factors for IPA diagnosis (Table 4).

## Differences in the incidence of IPA affected by COVID-19 control measures

In the current study, we found that COVID-19 infection was an independent risk factor for IPA. We further analyzed the possible changes in the incidence of IPA before and post-COVID-19 pandemic. As shown in Fig. 5A, we found the concordance rate of IPA diagnosis in post-COVID-19 patients was 47.62% (110/231), which was significantly higher than that in pre-COVID-19 patients (30.21% (29/96); $P$ = 0.004). Next, we analyzed the trend of *Aspergillus* cases throughout the entire study period and found that compared with the pre-COVID-19 period, the number of *Aspergillus* cases has been on the rise in the post-COVID-19 period (Fig. 5B). Additionally, there was no difference in the incidence of IPA between COVID-19 and non-COVID-19 patients (52.17% (36/69) vs 39.92% (103/258), $P$ = 0.067) (Fig. S2). Of all the positive cases with fungi infection, the strain of *Aspergillus fumigatus* was the most popular detected in pre-COVID-19 patients. To be noted, the strand of *Aspergillus terreus* and some other

**TABLE 4** Risk factors for IPA diagnosis[a]

| Variable | Univariate analysis | | Multivariate analysis | |
|---|---|---|---|---|
| | Or (95% CI) | P value | Or (95% CI) | P value |
| Age | 0.979 (0.964–0.995) | 0.010[**] | 0.987 (0.969–1.005) | 0.151 |
| Coma or drug sedation | 1.705 (1.012–2.871) | 0.045[*] | 1.794 (1.000–3.220) | 0.050 |
| Interstitial lung disease | 3.343 (1.472–7.591) | 0.004[**] | 3.439 (1.404–8.427) | 0.007[**] |
| Diabetes mellitus | 2.175 (1.295–3.654) | 0.003[**] | 1.934 (1.086–3.443) | 0.025[*] |
| Organ transplantation | 5.299 (1.450–19.374) | 0.012[*] | 4.817 (1.191–19.481) | 0.027[*] |
| Systemic corticosteroids | 2.771 (1.16–6.420) | 0.017[*] | 3.135 (1.251–7.852) | 0.015[*] |
| History of COVID-19 infection | 2.100 (1.266–3.485) | 0.004[**] | 2.120 (1.204–3.732) | 0.009[**] |
| Halo sign | 7.992 (1.742–36.665) | 0.007[**] | 5.460 (1.039–28.699) | 0.045[*] |
| Cavitation sign | 1.794 (1.001–3.216) | 0.050 | 1.696 (0.866–3.319) | 0.123 |
| Tracheobronchial ulcers, nodules, pseudomembranes, plaques, or eschars | 3.864 (1.937–7.708) | 0.000[***] | 3.012 (1.430–6.343) | 0.004[**] |

[a]*, P < 0.05; **, P < 0.01; ***, P < 0.001.

fungi, including *Rhizomucor*, *Rhizopus*, *Pneumocystis jirovecii*, *Candida*, and *Cryptococcus,* was also detected more frequently post-COVID-19 pandemic (Fig. 5D). Among the 139 IPA patients of our study, 27 cases of mixed infection were identified with other pathogens in the group of pre-COVID-19 patients. However, 107 cases of mixed infection with more kinds of co-infection patterns were found among patients post-COVID-19 pandemic (Fig. 5C and E). The numbers of cases with mixed infections and the predominant pathogens in IPA patients before and post-COVID-19 pandemic were summarized in Fig. S3. The results showed the highest positive rate of *Human gammaherpesvirus* (46.81% and 52.70%, respectively) before and post-COVID-19 pandemic. The other predominant pathogens were *Acinetobacter baumannii* (17.02%), *Stenotrophomonas maltophilia* (17.02%), *Pneumocystis jirovecii* (10.64%), and *Rhizomucor/Rhizopus* (8.51%) in pre-COVID-19 patients, and *Acinetobacter baumannii* (19.59%), *Klebsiella* (11.49%),

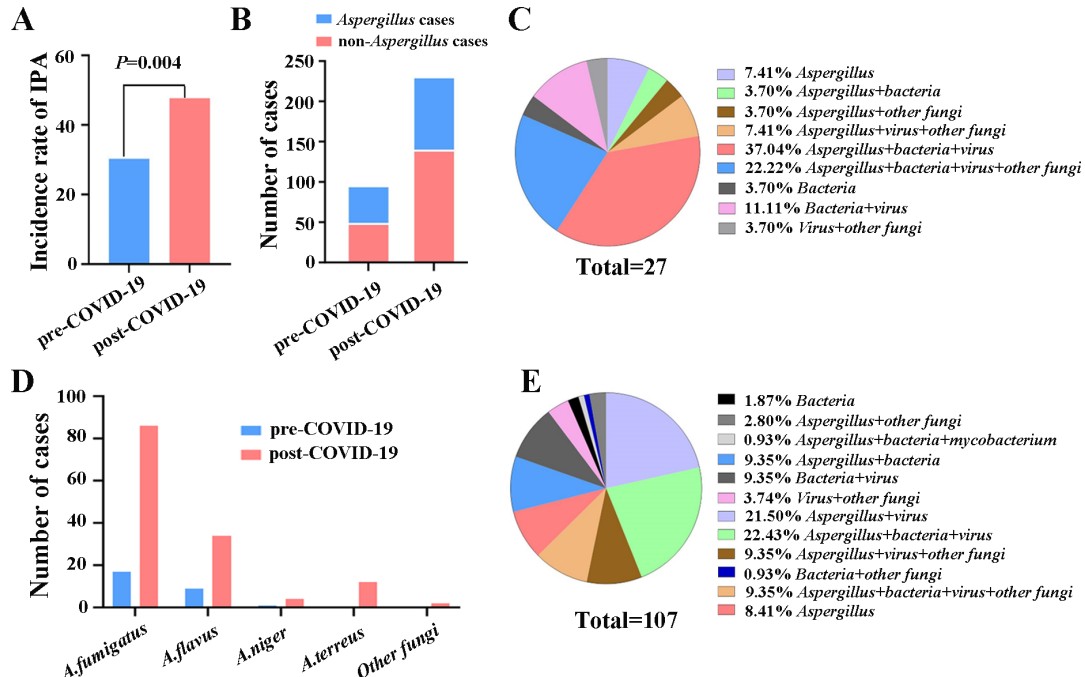

**FIG 5** Changes affected by COVID-19 control measures. (A) The occurrence rate of IPA before and post-COVID-19 pandemic. (B) The number of *Aspergillus* cases in pre-COVID-19 and post-COVID-19 group. (C) The patterns of mixed infections in the group of pre-COVID-19 patients. (D) The comparison of different fungi detected in the pre-COVID-19 and post-COVID-19 groups. (E) The patterns of mixed infections in patients post-COVID-19 pandemic.

*Rhizomucor/Rhizopus* (8.78%), and *Pseudomonas aeruginosa* (7.43%) in patients post-COVID-19 pandemic, respectively (Fig. S3).

## DISCUSSION

In recent years, the prevalence of pulmonary fungal disease has been on the rise. Especially, invasive pulmonary aspergillosis (IPA) is emerging as an important cause of mortality in patients with pulmonary fungal infections (19). For patients with IPA, early recognition and timely diagnosis can improve their clinical outcome. Unfortunately, the diagnosis of IPA is challenging due to the lack of specific clinical features, making etiological evidence crucial for diagnosis. Although relying on culture to diagnose fungal infection might delay initiation of treatment, because of the low sensitivity and false negatives of culture, positive culture of *Aspergillus* in biopsy was recommended for the final diagnosis of IPA (23). Additionally, empirical antifungal therapy is increasingly common in high-risk groups with suspected *Aspergillus* infection, resulting in low pathogen detection rates of traditional tests (24). Therefore, it is an urgent situation to explore new sensitive technology to improve *Aspergillus* identification.

The application of non-biopsied markers (including serum/BALF GM tests, serum B-D-glucan, and *Aspergillus* IgG test) and molecular biological methods (such as mNGS, *Aspergillus* polymerase chain reaction) could provide more options for the identification of pulmonary *Aspergillus* infection (25, 26). Previous studies showed that the BALF GM test was more efficient for IPA than serum GM (27, 28). A recent study found that the sensitivity and specificity of BALF GM (≥0.717) could reach up to 66.67% and 85.45%, respectively (29). In our study, we revealed that the sensitivity of BALF GM (≥1.0) test (range from 46.75% to 63.75%) outperformed serum GM (range from 19.65% to 34.90%), which was in accordance with other previous studies (27, 28). Antifungal treatment and neutrophil counts easily affect the serum GM test. We found that the sensitivity of serum GM was extremely low (only 26.62%) in these non-neutropenic patients with IPA. The reason may be that GM antigens rarely enter the peripheral blood in non-neutropenic IPA, characterized by avascular invasion (30). In addition, *Aspergillus* antibodies in blood can bind directly to GM antigens, which further lowers the level of serum GM (31).

In previous studies, mNGS has been widely used as a promising tool for the diagnosis of infectious diseases. The clinical experience of mNGS in diagnosing IPA is largely based on case reports and small sample studies. Miao et al. suggested that mNGS exhibited better performance than culture regarding fungal detection, but there was no significant difference in *Aspergillus* detection (OR, 3.7, 95% CI: 0.97–20.5; *P* = 0.057) due to the small sample size of IPA (26 cases) (32). Similarly, two recent studies with small samples showed that mNGS exhibited excellent performance in detecting *Aspergillus* (33) and revealed that the sensitivity of BALF mNGS was 92.31% for diagnosing IPA in non-neutropenic patients, significantly higher than other detection assays (29). Therefore, it is necessary to verify these preliminary conclusions with larger studies. In this study of 327 patients with suspected *Aspergillus* infection, we found mNGS exhibited excellent performance in detecting *Aspergillus*. Its sensitivity and NPV outperformed any routine microbiological test and can be comparable to the combination of multiple CMTs. These results indicated that mNGS acts as a useful culture-free assay in the diagnosis of IPA.

Serum GM test might cause false-negative results, especially in non-neutropenic patients, while BALF GM test is consistently recommended across recent consensus guidelines (34, 35). However, the diagnostic performance for IPA between BALF GM and mNGS is controversial. Ao et al. reported that BALF GM (57.7%) had the highest sensitivity in detecting *Aspergillus*, followed by mNGS (42.3%), culture (30.8%), and smear (7.7%) in 26 IPA patients (36). In contrast, another study observed improved sensitivity of mNGS compared with BALF GM (33), which was in accordance with our findings. A previous study found no significance of GM in the serum between the IPA and non-IPA groups (29), but our study showed a statistically significant difference in the level of serum/BALF GM between these two groups. These divergences mentioned above may

be attributed to differences in study population, sampling methods, GM cutoff value, and empirical antifungal therapy.

As is well known, accurate identification of strains is very important in guiding antifungal treatment. Unlike traditional tests, mNGS allows thousands to billions of DNA fragments to be independently sequenced simultaneously and is not influenced by genomic mutations or diversity (37). Among the 139 IPA patients of our study, approximately 45.32% (63/139) were positive only in mNGS, which was extremely higher than in traditional tests. These reminded us mNGS offers a more accurate analysis of pathogenic strains, which is even more specific than other methods. With the prevalence of prophylactic antifungal therapy in high-risk populations, it will lead to the emergence of azole-resistant strains. More and more physicians are relying on the rapid and accurate identification of pathogenic strains and even information on virulence and resistance provided by mNGS to guide antifungal therapy. Zhang et al. highlighted the favorable prognosis attributed to antifungal drug changes, which was based on mNGS results (38). We found mNGS results caused a direct shift in management of 212 (64.8%) positive effect patients, making a clear diagnosis and instructing antifungal therapy. Of these patients, 73 cases (22.3%) received voriconazole therapy based on mNGS results, while 40 cases (12.2%) received combination antifungal therapy. This suggested that mNGS might play a role in guiding targeted therapy in IPA patients. However, despite the strict algorithms incorporated into the bioanalysis software to rule out potential contamination, mNGS consequences in 11 patients were considered as colonization or contamination. In 104 (31.8%) patients, mNGS failed to identify any *Aspergillus*. We must admit that there are some challenges encountered when analyzing metagenomic data. First, the predominance of host-derived nucleic acids in clinical samples (e.g., blood, respiratory secretions) often obscures pathogen-derived sequences, reducing detection sensitivity. In addition, the difficulty of extracting nucleic acids varied among different species. Inadequate wall-breaking treatment might cause false-negative results. Second, the inherent complexity of mNGS data sets, combined with variable sequencing depths, posed significant challenges for identifying low-abundance pathogens. Even to this day, there is still a lack of widely accepted standards and quantitative thresholds. Third, the accuracy of mNGS was prone to be affected by nucleic acid contamination in background microbiome, reagents, and consumables. It was difficult to distinguish between pathogen infection and colonization as well as external nucleic acid sources. These findings suggested that it was essential to make the utilization of both conventional microbial tests and mNGS to increase diagnostic precision. Although there were many challenges in the clinical application of mNGS, we still found that mNGS diagnosis had a positive impact on patient prognosis. In any case, mNGS has the potential to revolutionize the management of fungal infections and is important to help physicians to formulate possible preventive strategies in time. Moreover, critical consideration should be given to mNGS results corroborated by host factors, imaging findings, and other microbiological evidence in clinic.

Furthermore, our study also revealed that most patients had mixed infections of *Aspergillus* and other pathogens, particularly "uncultured" *Human herpesvirus* and "indistinguishable" *Rhizomucor/Rhizopus*. Early identification of mixed infections remains a huge challenge. Traditional methods rely on pre-evaluating potential pathogens for targeted detection. These methods are highly susceptible to the subjective experience of clinical personnel and have a narrow spectrum of pathogens, leading to high rates of missed infections. In contrast, mNGS provides an unbiased sampling method that allows the simultaneous identification of all potentially infectious agents in a single run (33). This may explain the satisfactory performance of mNGS in identifying *Aspergillus* co-infections in this study. We firmly believe that the mNGS approach will not only simplify diagnosis methods of co-infections but also bring the greatest benefits to IPA patients who often suffer from complex infections.

In our study, we found that IPA was associated with severe clinical symptoms and poor outcomes. IPA patients had a higher incidence of coma or drug sedation, and

the 28-day mortality was higher than that of the control group. Notably, in addition to the common risk factors, such as interstitial lung disease, diabetes, and organ transplantation, we also found that patients with a history of COVID-19 infection were more prone to invasive pulmonary aspergillosis. The concordance rate of IPA diagnosis in post-COVID-19 patients was 47.62%, which was significantly higher than that in pre-COVID-19 patients (30.21%). It was alarming given the sheer number of people affected by COVID-19 infection. Although the COVID-19 pandemic is currently waning, seasonal epidemics persist with humans for a long time, and other respiratory viruses may also emerge in turn and pose the potential risks of IPA occurrence. In fact, repetitive damage and repair of alveolar tissues increase oxidative stress, inflammation, and elevated production of fibrotic proteins, ultimately disrupting normal lung physiology and skewing the balance towards the fibrotic milieu (39). All of these create favorable conditions for the growth of *Aspergillus*. In our study, the history of COVID-19 infection is also an independent risk factor for IPA. Identification of risk factors for IPA may raise clinical attention and require careful follow-up of high-risk individuals post-COVID-19 infection. The direct viral mechanism involved in the induction of pulmonary interstitial changes in recovered COVID-19 patients requires further investigation.

This study had some limitations. First, it was a single-center retrospective study; therefore, intrinsic bias was unavoidable. Second, clinical comprehensive diagnostic criteria were used as the gold standard, but it must be admitted that even experienced doctors cannot completely avoid clinical misdiagnosis, leading to biases in the diagnostic effects between different methods. Third, it was difficult to distinguish *Aspergillus* infection from colonization and contamination. In the future, the optimization and standardization of sampling and mNGS analysis might be a necessary condition to improve the diagnostic capability of genetic diagnosis.

In summary, mNGS is a feasible and highly sensitive method for detecting *Aspergillus* in suspected IPA patients. It also had the potential to revolutionize the management of fungal infections. However, mNGS couldn't yet replace CMTs. It was essential to make the utilization of both CMTs and mNGS to increase diagnostic precision. We should pay more attention to the incidence and risk factors for *Aspergillus* infection occurring post-COVID-19, and the interpretation of mNGS results should be combined with the actual condition of patients.

## ACKNOWLEDGMENTS

The authors wish to thank all research staff and patients for participating in this study.

This research received no external funding.

## AUTHOR AFFILIATIONS

[1]Department of Clinical Laboratory, Qilu Hospital of Shandong University, Jinan, Shandong, China
[2]Shandong Engineering Research Center of Biomarker and Artificial Intelligence Application, Jinan, Shandong, China
[3]Jinan AXZE Medical Test Laboratory, Jinan, Shandong, China
[4]Department of Infectious Diseases, Qilu Hospital of Shandong University, Jinan, Shandong, China

## AUTHOR ORCIDs

Lili Wang  http://orcid.org/0000-0002-7778-0768

## AUTHOR CONTRIBUTIONS

Ping Li, Conceptualization, Formal analysis, Software, Writing – original draft | Yuxia Chen, Formal analysis, Methodology | Xuelei Cao, Resources | Zheying Wang, Resources | Lintao Sai, Supervision | Lili Wang, Conceptualization, Supervision

## DATA AVAILABILITY

The raw sequence data reported in this paper have been deposited in the Genome Sequence Archive (24) in the National Genomics Data Center (Nucleic Acids Res 2022), China National Center for Bioinformation/Beijing Institute of Genomics, Chinese Academy of Sciences (GSA: CRA023528) and are publicly accessible at https://bigd.big.ac.cn/gsa/browse/CRA023528.

## ETHICS APPROVAL

The study was conducted in accordance with the amended Declaration of Helsinki. The human ethics and consent to participate was provided by the ethics committee of Qilu Hospital of Shandong University (KYLL-2021-120).

## ADDITIONAL FILES

The following material is available online.

### Supplemental Material

**Figure S1 (Spectrum00121-25-s0001.tif).** Overall survival analysis for mNGS (+) and mNGS (-) patients.
**Figure S2 (Spectrum00121-25-s0002.tif).** The incidence of IPA in COVID-19 and non-COVID-19 groups.
**Figure S3 (Spectrum00121-25-s0003.tif).** Mixed infections and the predominant pathogens identified in IPA patients before and postCOVID-19 pandemic.
**Supplemental material (Spectrum00121-25-s0004.docx).** Additional experimental details.

### Open Peer Review

**PEER REVIEW HISTORY (review-history.pdf).** An accounting of the reviewer comments and feedback.

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
