## [Reviewer comments · Microbiology Spectrum]

Microbiology Spectrum

Diagnosis of invasive pulmonary aspergillosis using Metagenomic Next-Generation Sequencing and conventional microbial tests post COVID-19 pandemic

Ping Li, Yuxia Chen, Xuelei Cao, Zheyang Wang, Lintao Sai, and Lili Wang

Corresponding Author(s): Lili Wang, Department of Clinical Laboratory, Qilu Hospital of Shandong University

Review Timeline:

Submission Date:	January 13, 2025
Editorial Decision:	February 10, 2025
Revision Received:	April 10, 2025
Accepted:	April 29, 2025

Editor: Dhammika Navarathna

Reviewer(s): The reviewers have opted to remain anonymous.

Transaction Report:

DOI: <https://doi.org/10.1128/spectrum.00121-25>

Re: Spectrum00121-25 (Diagnosis of invasive pulmonary aspergillosis using Metagenomic Next-Generation Sequencing and conventional microbial tests post COVID-19 pandemic)

Dear Dr. Lili Wang:

Thank you for the privilege of reviewing your work. Below you will find my comments, instructions from the Spectrum editorial office, and the reviewer comments.

Please note that public repository of mNGS data from BALF specimens are required and clarify the time period defined as pre- and post-COVID pandemic

Revision Guidelines

Sincerely,
Dhammika Navarathna
Editor
Microbiology Spectrum

Reviewer #1 (Comments for the Author):

1. Include explanation for host factors for diagnostic criteria for IPA. 77 of non-IPA group subjects were both culture and mNGS positive for fungi (Figure-1). Are these subjects not under IPA group based on no host factors?
2. One of the main concerns in the field of mNGS as diagnostic tool is false positive rate and the impact on antimicrobial

stewardship. According to the study, there were 212 patients with positive clinical change or antifungal therapy. How many of these patients belong to IPA vs non-IPA group? This information is an important aspect as specificity of mNGS assay is relatively low. Incorporate IPA and Non-IPA patient numbers into table 3.

3. Was there any positive impact on mNGS diagnostics in patient outcomes (table1)?

Reviewer #2 (Comments for the Author):

Review Spectrum00121-25

Invasive pulmonary aspergillosis (IPA) predominantly affects individuals with compromised immune systems, and is associated with a poor prognosis and a high mortality rate, which can exceed 50% in neutropenic patients. Diagnosing IPA through phenotypic methods can be challenging. As a result, molecular techniques have become increasingly utilized for more accurate diagnosis. The authors conducted a retrospective study to compare the effectiveness of metagenomic and traditional diagnostic methods for diagnosis of IPA. Additionally, they investigated the incidence and risk factors associated with *Aspergillus* infections occurring post COVID-19. Below are my comments- Please include line numbers in the document, as it was quite challenging to review without them.

Major comment:

The study focused on IPA during the post-COVID-19 period as the title indicates. In the methods section, it would be helpful to specify the exact time frame defined as "before and after the COVID-19 pandemic," particularly for the period from October 2021 to June 2023. It would also be useful to clarify when the data for the pre-COVID period was collected.

Regarding Figure 5A: It would be more informative to display the trend of *Aspergillus* cases throughout the entire study period, including both the pre- and post-COVID phases. Additionally, it would be clearer to separate and indicate the incidence of IPA in COVID and non-COVID patients, rather than simply showing the incidence in COVID and non-COVID groups during an unspecified period.

Minor Comments:

Abstract:

- Line 19: Please provide the full name for *A. baumannii*.

Study Design and Subjects:

- Line 18: Remove the ethics declaration, as it is already covered under ethical approval and consent.

Diagnostic Criteria for IPA:

- Line 6: Change the future tense to past tense (e.g., "condition was met" instead of "will be met").

Microbiological Testing and mNGS:

- Under the supplementary section, include a detailed method describing how culture, smear microscopy, and the GM test were performed. For example, specify the volume of BALF cultured, the type of media used, incubation times, and temperatures.

- Line 5: Specify whether you used peripheral blood or serum.

- Line 7: Clarify from which specimen type nucleic acid was extracted.

- Line 8: Use "are" instead of "were."

Supplementary:

- Should it be "RNA enrichment" or "enrichmen"?

Figure 2B:

- Please add the results for other *Aspergillus* species to the graph.

Clinical Effects of mNGS Results on IPA Diagnosis and Management:

- Line 8: Italicize *Aspergillus*.

- Line 8 and Table 3: Is the anti-*Aspergillus* treatment different from voriconazole? If so, please define it.

- Line 9: "11 cases were regarded as false positive"-what criteria were used to determine this?

Mixed Infections and Co-pathogens Detected by mNGS in IPA Patients:

- Line 7: Ensure that you provide the full name for all pathogens, especially the first time they are mentioned.

Difference in the Incidence of IPA:

- Correct the spelling of "CIVIVD-19" to "COVID-19 control measures."

- Line 3: Again, specify the period for "before and post-COVID pandemic" in the methods section.

- Line 8: "Cryptococcus" should start with a capital letter.

- Line 20: Provide the full name for *P. aeruginosa*.

Discussion:

- Line 15: Italicize *Aspergillus*.

- Line 48: Italicize *Aspergillus*.

- Line 72: On what basis was it considered colonization or contamination?

It would also be useful to discuss any challenges encountered when analyzing metagenomic data

Table 2:

- Correct "Methords" to "Methods."

Table 3:

- Include the name of the previous antifungal therapy regimen.

Figure 1:

- Include the results for the mNGS/culture tests in the 139 IPA group, as you did for the 188 non-IPA group.

Figure 2A:

- Instead of "double," specify the exact test (e.g., mNGS & traditional tests).

Figure 2B:

- Add the results for other *Aspergillus* species to the graph.

Figure 2:

- After "comparison," add "of 157 *Aspergillus* strains detected."

Figure 4:

- Write the full names of the organisms.

Review Spectrum00121-25

Invasive pulmonary aspergillosis (IPA) predominantly affects individuals with compromised immune systems, and is associated with a poor prognosis and a high mortality rate, which can exceed 50% in neutropenic patients. Diagnosing IPA through phenotypic methods can be challenging. As a result, molecular techniques have become increasingly utilized for more accurate diagnosis. The authors conducted a retrospective study to compare the effectiveness of metagenomic and traditional diagnostic methods for diagnosis of IPA. Additionally, they investigated the incidence and risk factors associated with *Aspergillus* infections occurring post COVID-19. Below are my comments- Please include line numbers in the document, as it was quite challenging to review without them.

Major comment:

The study focused on IPA during the post-COVID-19 period as the title indicates. In the methods section, it would be helpful to specify the exact time frame defined as "before and after the COVID-19 pandemic," particularly for the period from October 2021 to June 2023. It would also be useful to clarify when the data for the pre-COVID period was collected.

Regarding Figure 5A: It would be more informative to display the trend of *Aspergillus* cases throughout the entire study period, including both the pre- and post-COVID phases. Additionally, it would be clearer to separate and indicate the incidence of IPA in COVID and non-COVID patients, rather than simply showing the incidence in COVID and non-COVID groups during an unspecified period.

Minor Comments:

Abstract:

- Line 19: Please provide the full name for *A. baumannii*.

Study Design and Subjects:

- Line 18: Remove the ethics declaration, as it is already covered under ethical approval and consent.

Diagnostic Criteria for IPA:

- Line 6: Change the future tense to past tense (e.g., "condition was met" instead of "will be met").

Microbiological Testing and mNGS:

- Under the supplementary section, include a detailed method describing how culture, smear microscopy, and the GM test were performed. For example, specify the volume of BALF cultured, the type of media used, incubation times, and temperatures.
- Line 5: Specify whether you used peripheral blood or serum.
- Line 7: Clarify from which specimen type nucleic acid was extracted.
- Line 8: Use "are" instead of "were."

Supplementary:

- Should it be "RNA enrichment" or "enrichmen"?

Figure 2B:

- Please add the results for other *Aspergillus* species to the graph.

Clinical Effects of mNGS Results on IPA Diagnosis and Management:

- Line 8: Italicize *Aspergillus*.
- Line 8 and Table 3: Is the anti-*Aspergillus* treatment different from voriconazole? If so, please define it.

- Line 9: "11 cases were regarded as false positive"—what criteria were used to determine this?

Mixed Infections and Co-pathogens Detected by mNGS in IPA Patients:

- Line 7: Ensure that you provide the full name for all pathogens, especially the first time they are mentioned.

Difference in the Incidence of IPA:

- Correct the spelling of "CIVIVD-19" to "COVID-19 control measures."
- Line 3: Again, specify the period for "before and post-COVID pandemic" in the methods section.
- Line 8: "*Cryptococcus*" should start with a capital letter.
- Line 20: Provide the full name for *P. aeruginosa*.

Discussion:

- Line 15: Italicize *Aspergillus*.
- Line 48: Italicize *Aspergillus*.
- Line 72: On what basis was it considered colonization or contamination?

Table 2:

- Correct "Methords" to "Methods."

Table 3:

- Include the name of the previous antifungal therapy regimen.

Figure 1:

- Include the results for the mNGS/culture tests in the 139 IPA group, as you did for the 188 non-IPA group.

Figure 2A:

- Instead of "double," specify the exact test (e.g., mNGS & traditional tests).

Figure 2B:

- Add the results for other *Aspergillus* species to the graph.

Figure 2:

- After "comparison," add "of 157 *Aspergillus* strains detected."

Figure 4:

- Write the full names of the organisms.

Reviewer comments

Reviewer #1:

1. Include explanation for host factors for diagnostic criteria for IPA. 77 of non-IPA group subjects were both culture and mNGS positive for fungi (Figure-1). Are these subjects not under IPA group based on no host factors?

Response: Thank you for your critical comment and we have included a detailed explanation for host factors for diagnostic criteria for IPA in the section of “Diagnostic criteria for IPA”. The host factors for diagnostic criteria for IPA were according to the guidelines of 2020 European Organization for Research and Treatment of Cancer and the Mycoses Study Group Education and Research Consortium (EORTC/MSGERC). All the 77 subjects were not under IPA group based on no host factors.

2. One of the main concerns in the field of mNGS as diagnostic tool is false positive rate and the impact on antimicrobial stewardship. According to the study, there were 212 patients with positive clinical change or antifungal therapy. How many of these patients belong to IPA vs non-IPA group? This information is an important aspect as specificity of mNGS assay is relatively low. Incorporate IPA and Non-IPA patient numbers into table 3.

Response: Thank you for your kind comment. We fully agree with your opinions and have systematically re-checked clinical data based on treatment regimen modification and revised the manuscript as you suggested. The numbers of IPA and non-IPA patients have been incorporated into Table 3. Of the 212 patients with positive clinical change or antifungal therapy, 113 patients were in the IPA group and 99 patients were in the non-IPA group. More details were shown in the section of “Clinical effects of mNGS results on IPA diagnosis and management”. In addition, we are also fully aware the issue of false positive rate and low specificity in mNGS, which may be affected by environmental contamination, the potential interference caused by strongly positive samples or other accidentally factors. And the patients with non-IPA

may include colonizing microbes or non-invasive infection. In clinical practice, critical consideration should be given to mNGS results corroborating by host factors, imaging findings, and other microbiological evidence. In the section of “Discussion”, we have analyzed some keypoints about mNGS roles in guiding treatment regimen and modified the manuscript with more information about the mNGS challenges in clinical application.

3. Was there any positive impact on mNGS diagnostics in patient outcomes (table1)?

Response: Thank you for your comments and we have modified the manuscript in the section of “Clinical effects of mNGS results on IPA diagnosis and management”. The patients were divided into two groups based on the detection of *Aspergillus* by mNGS: mNGS (+) group and mNGS (-) group, containing 189 and 138 patients, respectively. The 28-days overall survival (OS) in mNGS (+) group was significantly lower than that in mNGS (-) group ($p=0.018$) (Figure S1), indicating that mNGS diagnosis might play the role in suggesting a poor prognosis for patients and helping doctors to formulate possible preventive strategies in time. Thank you again.

Reviewer #2:

Invasive pulmonary aspergillosis (IPA) predominantly affects individuals with compromised immune systems, and is associated with a poor prognosis and a high mortality rate, which can exceed 50% in neutropenic patients. Diagnosing IPA through phenotypic methods can be challenging. As a result, molecular techniques have become increasingly utilized for more accurate diagnosis. The authors conducted a retrospective study to compare the effectiveness of metagenomic and traditional diagnostic methods for diagnosis of IPA. Additionally, they investigated the incidence and risk factors associated with *Aspergillus* infections occurring post COVID-19. Below are my comments- Please include line numbers in the document, as it was quite challenging to review without them.

1. The study focused on IPA during the post-COVID-19 period as the title indicates. In the methods section, it would be helpful to specify the exact time frame defined as

"before and after the COVID-19 pandemic," particularly for the period from October 2021 to June 2023. It would also be useful to clarify when the data for the pre-COVID period was collected.

Response: Thank you for your kind suggestion and so sorry about the unclear time frame "before and after the COVID-19 pandemic". We have modified the manuscript as you suggested (Line 103-106). The epidemic prevention and control policy of COVID-19 was gradually eased since December 7, 2022 in China. Before that, due to the strict epidemic prevention and control policies, there were no patients with COVID-19 infection visiting Qilu Hospital. So, taking December 7, 2022 as the cut-off point in the current study, the timeframe from October 1, 2021 to December 7, 2022 was defined as the pre-COVID-19 pandemic era, while December 8, 2022 to June 30, 2023 was categorized as the post-COVID-19 pandemic era.

Major comment:

Regarding Figure 5A: It would be more informative to display the trend of *Aspergillus* cases throughout the entire study period, including both the pre- and post-COVID phases. Additionally, it would be clearer to separate and indicate the incidence of IPA in COVID and non-COVID patients, rather than simply showing the incidence in COVID and non-COVID groups during an unspecified period.

Response: Thank you for your comment. We have displayed the trend of *Aspergillus* cases throughout the entire study period in Figure 5B. We found that compared with the pre-COVID-19 period, the number of *Aspergillus* cases has been on the rise in the post-COVID-19 period. Then, we analyzed the incidence of IPA in COVID and non-COVID patients. However, we found there was no difference in the incidence of IPA between COVID-19 and non-COVID-19 patients (52.17% (36/69) vs 39.92% (103/258), $P=0.067$) (Figure S2). The manuscript was also modified in the section of "Differences in the incidence of IPA affected by COVID-19 control measures" (Line 293-300).

2. Minor Comments:

Abstract:

- Line 19: Please provide the full name for *A. baumannii*.

Response: Thanks for your correction. We have provided the full name for *A. baumannii* (Line 35).

Study Design and Subjects:

- Line 18: Remove the ethics declaration, as it is already covered under ethical approval and consent.

Response: We have removed the ethics declaration.

Diagnostic Criteria for IPA:

- Line 6: Change the future tense to past tense (e.g., "condition was met" instead of "will be met").

Response: Thanks. We have changed the "condition was met" to "will be met" (Line 148).

Microbiological Testing and mNGS:

- Under the supplementary section, include a detailed method describing how culture, smear microscopy, and the GM test were performed. For example, specify the volume of BALF cultured, the type of media used, incubation times, and temperatures.

Response: We have added detailed experimental procedures for culture, smear microscopy, and the GM test to the supplementary section (Supplement/Line 19-42).

- Line 5: Specify whether you used peripheral blood or serum.

Response: We used serum for GM test and we have corrected that in the manuscript (Line 164).

- Line 7: Clarify from which specimen type nucleic acid was extracted.

Response: Deoxyribonucleic acid was extracted in our study (Line 166).

- Line 8: Use "are" instead of "were."

Response: We have changed "were" to "are" (Line 168). Thank you.

Supplementary:

- Should it be "RNA enrichment" or "enrichmen"?

Response: Thanks for your comment. The current study analyzed the data of the DNA sequencing process, we have deleted the "RNA enrichment" and "Reverse transcription and two-strand synthesis" in the supplementary section. Sorry for our carelessness.

Figure 2B:

- Please add the results for other *Aspergillus* species to the graph.

Response: We have added the results for other *Aspergillus* species to the graph. The manuscript was also modified (Line 214).

Clinical Effects of mNGS Results on IPA Diagnosis and Management:

- Line 8: Italicize *Aspergillus*.

Response: We have italicized *Aspergillus* (Line 260).

- Line 8 and Table 3: Is the anti-*Aspergillus* treatment different from voriconazole? If so, please define it.

Response: Thank you for your kind suggestion and sorry for the unclear expression with the treatment for anti-*Aspergillus*. Because voriconazole is the first-line treatment for *Aspergillus* infections, empirical anti-*Aspergillus* treatment refers to the use of voriconazole for treating *Aspergillus* infections in our paper. We have changed "empirical anti-*Aspergillus* therapy" to "empirical voriconazole therapy" and modified the manuscript (Line 260) and Table 3.

- Line 9: "11 cases were regarded as false positive"-what criteria were used to determine this?

Response: Thanks for your comment. The final determination of causative pathogens,

colonization or contamination was based on clinical comprehensive diagnostic criteria, which was established by two senior specialists after independently reviewed the electronic medical records of each patient. We have made the corresponding explanation in the section of Materials and Methods (Line 153-158).

Mixed Infections and Co-pathogens Detected by mNGS in IPA Patients:

- Line 7: Ensure that you provide the full name for all pathogens, especially the first time they are mentioned.

Response: We have checked and provided the full name for all pathogens (Line 278-279).

Difference in the Incidence of IPA:

- Correct the spelling of "CIVIVD-19" to "COVID-19 control measures."

Response: We have corrected the spelling of "CIVIVD-19" to "COVID-19" (Line 298).

- Line 3: Again, specify the period for "before and post-COVID pandemic" in the methods section.

Response: We have specified the period for "before and post-COVID pandemic" in the methods section (Line 114-117). Thank you for your suggestion.

- Line 8: "Cryptococcus" should start with a capital letter.

Response: We have changed cryptococcus to *Cryptococcus* (Line 315).

- Line 20: Provide the full name for *P. aeruginosa*.

Response: We have provided the full name for *P. aeruginosa* (Line 329).

Discussion:

- Line 15: Italicize *Aspergillus*.

Response: We have italicized *Aspergillus* (Line 346).

- Line 48: Italicize *Aspergillus*.

Response: We have italicized *Aspergillus* (Line 379).

- Line 72: On what basis was it considered colonization or contamination?

It would also be useful to discuss any challenges encountered when analyzing metagenomic data.

Response: Thank you for your kind suggestion. The final determination of colonization or contamination was based on clinical comprehensive diagnostic criteria, which was established by two senior specialists after independently reviewed the electronic medical records of each patient, based on clinical symptoms, radiologic features, microbiologic tests and treatment responses. We have made the corresponding explanation in the section of “Materials and Methods” (Line 153-158) and “Discussion” (Line 407-418).

Table 2:

- Correct "Methodors" to "Methods."

Response: We have corrected "Methodors" to "Methods" in Table 2.

Table 3:

- Include the name of the previous antifungal therapy regimen.

Response: Thank you for your comment. The Changes in antifungal therapy regimen have been included in Table3.

Figure 1:

- Include the results for the mNGS/culture tests in the 139 IPA group, as you did for the 188 non-IPA group.

Response: We have added the results for the mNGS/culture tests in the 139 IPA group, as we did for the 188 non-IPA group in Figure 1.

Figure 2A:

- Instead of "double," specify the exact test (e.g., mNGS & traditional tests).

Response: We have changed "double" to " mNGS & traditional tests " in Figure 2A.

Figure 2B:

- Add the results for other *Aspergillus* species to the graph.

Response: We have added the results for other *Aspergillus* species to the graph.

Figure 2:

- After "comparison," add "of 157 *Aspergillus* strains detected."

Response: We have added "of 157 *Aspergillus* strains detected" after "comparison" in Figure 2 legend.

Figure 4:

- Write the full names of the organisms.

Response: We have written the full names of the organisms in Figure 4. Thanks for your kind suggestion.

Re: Spectrum00121-25R1 (Diagnosis of invasive pulmonary aspergillosis using Metagenomic Next-Generation Sequencing and conventional microbial tests post COVID-19 pandemic)

Dear Dr. Lili Wang:

Your manuscript has been accepted, and I am forwarding it to the ASM production staff for publication. Your paper will first be checked to make sure all elements meet the technical requirements. ASM staff will contact you if anything needs to be revised before copyediting and production can begin. Otherwise, you will be notified when your proofs are ready to be viewed.

Sincerely,
Dhammika Navarathna
Editor
Microbiology Spectrum

Reviewer #1 (Comments for the Author):

Revision has been completed as suggested by reviews

Reviewer #2 (Comments for the Author):

Thanks for addressing the comments